# Longitudinal Changes in Occupational Balance among Baby Boomers in Japan (1996–2016)

**DOI:** 10.3390/ijerph20054060

**Published:** 2023-02-24

**Authors:** Makoto Watanabe

**Affiliations:** School of Allied Health Sciences, Kitasato University, 1-15-1 Kitazato, Minami-Ku, Sagamihara 252-0373, Japan; mwat@ahs.kitasato-u.ac.jp

**Keywords:** occupational balance, the older adult population, readjustment, role changes, role overload

## Abstract

In Japan, the proportion of the older adult population—the post-war baby boomer generation—is increasing rapidly and presenting new problems, such as suicide among baby boomers and the burden of family care. The purpose of this study was to clarify how baby boomers changed their occupational balance between their 40s and 60s. This study used public statistical data from the Survey on Time Use and Leisure Activities published by the Statistics Bureau of Japan to analyze the longitudinal characteristics of the time allocation of baby boomers. The findings of this study showed gender differences in occupational balance among the study population. The occupational balance of men changed due to occupational transition after mandatory retirement, but that of women did not change significantly. Longitudinally examining the time allocation changes of one generation revealed that the readjustment of occupational balance is necessary during life role changes, such as retirement. Moreover, if this readjustment is not carried out properly, individuals will experience role overload and loss.

## 1. Introduction

### 1.1. Baby Boomer Issues in Japan

In Japan, the proportion of the older adult population (aged 65 and over) is increasing rapidly. In 2007, this population exceeded 21% of the total population, indicating that Japan had entered a so-called “super-aging society,” and this proportion is projected to reach approximately 30% of the total population by 2025. This phenomenon of an increase in the aging population is not unique to Japan but is occurring on a global scale [1].

To deal with this situation, the universal health insurance system has aimed to foster health promotion in Japan, and the long-term care insurance system is also reportedly effective [2]. Although the medical and welfare systems support the health status of older adults, the growth of this population presents new problems, such as higher suicide rates among older individuals and the burden of family care.

The main generation of adults aged 65 and older—the post-war baby boomer generation—drove Japan’s high economic growth after the war and is the most populous age group in Japan today. However, this generation finished reaching the age of retirement in 2015, and many are in the process of transitioning away from work. Thus, the new problems associated with the growth of this population may be caused by challenges in changing occupational balance and roles within the family due to retirement.

### 1.2. Theoretical Background

Since its inception, the field of occupational therapy has taken a great interest in occupational balance with a particular focus on daily habits. Meyer emphasized that work, play, rest, and sleep must be balanced even in difficult situations [3], and Slagle noted that “our lives are made up of habit reactions” [4]. Llorens also found that balance in the individual environment supports functional performance in occupational domains [5]. Moreover, the desired occupational balance—which contributes to health and well-being—consists of “meaningful activity” and “meaningful redundancy” [6]. Wagman et al. analyzed the concept of occupational balance, defining it as an individual’s perception that there is an appropriate amount of occupation and an appropriate variation between occupations [7].

Studies on occupational balance have been conducted on a variety of populations: for example, mothers with preschool children [8], cardiac patients [9], community-based adults with spinal cord injuries [10], probationary young offenders [11], and adults with rheumatoid arthritis [12,13]. Occupational balance research has also been carried out in the field of psychiatry. Leufstadius et al. investigated time use among people with mental illness [14], and other studies have examined occupational balance in the everyday lives of women with stress-related disorders [15,16]. Bejerholm et al. attempted to create profiles of occupational engagement in people with schizophrenia [17], and Edgelow and Krupa used these profiles to examine the effects of interventions for people with serious mental illness [18]. Additionally, the occupational lives of older adults [19] and residents of Alzheimer’s special care units [20] have been examined. Previous time-use studies in occupational therapy have limited the nature of the disorder and the duration of the analysis, but occupational balance research has evolved to explore its association with health and satisfaction [21]. 

### 1.3. Hypothesis Develoment

Previous studies quantitatively or qualitatively investigated occupational balance for limited periods and for particular groups, and few previous studies targeted healthy older people. Wagman et al. quantitatively surveyed 153 adults aged 18 and over in Sweden using the “Occupational Balance Questionnaire (OBQ)” and found that there was a significant positive correlation between health status and life satisfaction, but age and OBQ scores were not found to be significantly correlated [22]. Additionally, Park et al. examined the relationship between roles, work balance, and QOL in 90 urban elderly people in South Korea, and showed that there was a mutual correlation [23]. Hovbrandt et al. conducted a qualitative study based on interviews with workers aged 65 and over. Their findings suggested that having functional competencies and resources to complement those competencies, harmonious occupation combinations, and alignment with values and personal meanings contribute to participants’ occupational balance [24]. 

Therefore, in order to deepen the examination of the occupational balance, it is necessary to focus on the current elderly generation and examine it from a longitudinal perspective. One of methods of studying occupational balance is to capture its transition in a certain generation over time; the official statistical data in Japan include time spent on life activities, such as self-care, work, and housework. In addition, the surveys conducted by the national statistics bureaus are nationwide cross-sectional surveys, and the purpose of the surveys is to clarify the state of occupational balance at a certain point in time. By using the data for each year obtained from these cross-sectional surveys, it is possible to consider from a longitudinal perspective. Accordingly, an investigation using those longitudinal data would be appropriate to examine an occupational balance. In other words, by using those longitudinal data, it will be possible to clarify longitudinally how baby boomers have changed their time allocation between their 40s and 60s. 

Thus, based on the above considerations, we propose two hypotheses, as follows:

**Hypothesis** **1.**
*Over time, less working time leads to more time spent in social activities to supplement social participation.*


**Hypothesis** **2.**
*Over time, the occupational balance shifts into harmony, with time for social activities and time for activities that are meaningful to the individual.*


Another thing we need to consider from a different perspective is the effect of gender differences on occupational balance. Håkansson et al. administered a questionnaire to 488 middle-aged women in Sweden on perceptions of daily occupations, perceived control, and life satisfaction. This research found that women who felt more satisfied with their occupational balance and occupational value perceived greater life satisfaction than other women [25]. On the practical side, couples who care for both dependent children and aging parents constitute an under-studied group, the so-called ‘sandwich generation’ [26]. In particular, it is expected that women in this generation will be required to fulfill multiple roles, and it is necessary to consider the possibility of burnout due to stressful situations. How have the women of this baby boom generation changed their allocation of time for each activity over time? On the other hand, have men shifted to spend more time with their families, such as doing housework, as their working hours have decreased?

Additionally, we hypothesize the following:

**Hypothesis** **3.**
*Over time, women in this generation spend less time on IADLs, such as housework, and more time with family, such as leisure.*


**Hypothesis** **4.**
*Over time, men in this generation spend less time working and more time doing IADLs, such as housework, and spending more time on family activities, such as leisure.*


In summary, this study focuses on how baby boomers in Japan have formed their occupational balance and allocated time to each activity of their daily lives. The purpose of this study is to clarify how the occupational balance has been formed in this generation by longitudinal analysis and to examine its characteristics.

## 2. Materials and Methods

### 2.1. Study Designs

This study constructed secondary data based on the national survey database, and analyzed the longitudinal characteristics of the time allocation of baby boomers in Japan. In general, these official statistical data have been used in the field of economics to show the trend of people’s time allocation [27]. According to a review by Gross [28], “time allocation” has been practiced as a research method via time-use surveys in the field of cultural anthropology since the 1920s. Time-use surveys can help determine how an individual allocates their 24 h a day in small time units. 

Therefore, this study focused not on cross-cutting daily occupational balance but on longitudinal analysis of the baby boomer generation’s shifting occupational balance to clarify how they allocated and spent time on each life activity from their 40s through their 60s. In this study, we not only investigated changes over time in the amount of time spent on daily activities among baby boomers, but also focused on gender differences in occupational balance as one of the factors affecting time allocation. In addition, since it is necessary to consider the effects of retirement when examining changes over time, we examined the effects of the day of the week, such as weekdays and Sundays, on occupational balance.

### 2.2. The Public Survey Used in This Study

This study used the Survey on Time Use and Leisure Activities published by the Statistics Bureau of Japan [29], which aims to “obtain basic data for clarifying the actual state of social life of the people, such as the allocation of living time and the status of major activities in leisure time” [29] and has been conducted every 5 years since 1976. The survey targets approximately 200,000 individuals aged 10 and over from approximately 88,000 households randomly selected from those within the designated survey area (approximately 7300 survey areas nationwide). The statistical data summarize items related to the average time spent on daily living behaviors, the status of living behavior by time zone, and the average time of major living behaviors. 

Each life activity in the survey is categorized as a primary, secondary, or tertiary activity. Primary activities include sleep, personal affairs, and meals. Secondary activities include commuting to work or school, work, schoolwork, housework, long-term care and nursing, childcare, and shopping. The tertiary activities include movement; TV, radio, newspaper, and magazines; rest and relaxation; learning, self-development, and training; hobbies and entertainment; sports; volunteer activities and social participation activities; dating and socializing; and consultation and medical treatment, among others.

### 2.3. Procedures for Data Analysis (Figure 1)

The Survey on Time Use and Leisure Activities classifies daily activities into 20 types and investigates the status of activities by time zone (by 15-min units). These daily life activities are divided into three categories: physiologically necessary activities, such as sleep and meals; obligatory activities, such as work, housework, and social activities; and other activities in one’s free time.

**Figure 1 ijerph-20-04060-f001:**
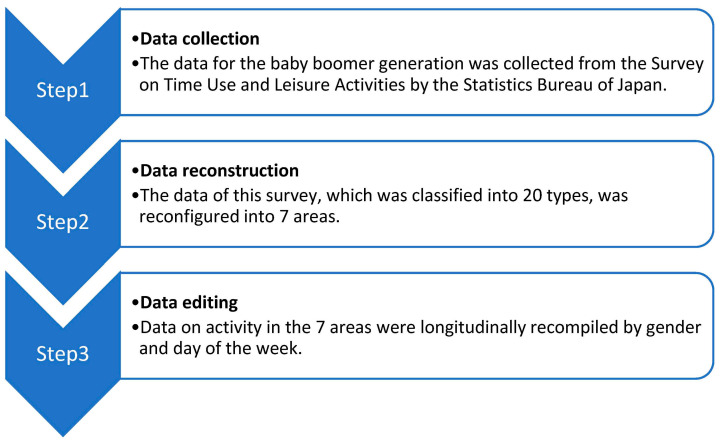
Study procedure.

In this study, data were collected and compiled into secondary data according to the procedures shown below.

First, data for the baby boomer generation were collected from the Survey on Time Use and Leisure Activities by the Statistics Bureau of Japan [29].

Second, in this survey, daily activities were classified into 20 types. Therefore, we referred to the basic framework of occupational therapy [30] (pp. 30–34) to classify each occupational activity of daily life into the following seven areas: (1) rest and sleep, (2) activities of daily living (ADLs), (3) instrumental activities of daily living (IADLs), (4) work, (5) leisure, (6) social participation, and (7) media viewing. Although media viewing is not included in the framework of occupational therapy, we added the category to this study because it is well established in Japan as a means of entertainment and information gathering.

Third, data on activity in the seven domains were longitudinally recompiled by gender and day of the week. 

## 3. Results

### 3.1. Characteristics of the Data Samples Used in This Study

The survey subjects were grouped by 5 years of age, and each living time was tabulated by attribute. For this study, we extracted data for the target generation over multiple survey years to analyze the baby boomers longitudinally.

The baby boomer generation (birth years 1947–1949) included individuals from 65 to 69 years of age (birth years 1947–1951) at the time of the survey conducted in 2016. At the time of data collection for this study, data from the 1996, 2001, 2006, 2011, and 2016 survey iterations were available. Therefore, the data used in this study were extracted from the age groups of 45–49 years in the 1996 survey, 50–54 years in the 2001 survey, 55–59 years in the 2006 survey, 60–64 years in the 2011 survey, and 65–69 years in the 2016 survey.

### 3.2. Longitudinal Characteristics of Time Allocation for Baby Boomers

Table 1 shows the temporal changes in the time allocation of baby boomers from 1996 to 2016. Time allocation trends for this generation show an increase in the amount of time required for basic activities such as rest, sleep, and ADLs, as opposed to a decrease in working hours (on weekdays, working hours for men decreased from 519 min to 224 min, and working hours for women decreased from 274 min to 108 min). Furthermore, media viewing (television, radio, newspapers, magazines, etc.) in particular is on the rise. These trends were almost the same for both men and women (on weekdays, media viewing time for men increased from 118 min to 228 min, while media usage time for women increased from 127 min to 191 min). However, time spent on social participation tended to decrease over time (on Sundays, men’s social participation time decreased from 38 min to 22 min, while women’s social participation time decreased from 32 min to 23 min).

### 3.3. Effects of Day of the Week on Occupational Patterns

Figure 2 shows changes in the time allocation of the baby boomer generation by survey year, divided into weekdays and Sundays, and shown as a ratio. There was a notable difference in working hours between weekdays and Sundays. Compared to weekdays, the proportion of hours worked on Sundays decreased. However, before age 65 (i.e., the age of retirement), the amount of time spent resting, sleeping, and viewing media on Sundays increased by about 10%.

Overall, there was no difference in age in the distribution of time spent on Sundays. Furthermore, there was no significant difference in the time required for IADLs between weekdays and Sundays.

### 3.4. Effects of Gender on Occupational Patterns

A characteristic of Japan’s baby boomer generation is that men and women spend different amounts of time on work and housework (see Table 1 and Figure 2); particularly, women spend more time on housework than men. This trend was observed not only on weekdays but also on Sundays when working hours were reduced. Remarkably, men spent twice as much leisure time as women on Sundays, and there was no significant change over time.

Figure 3 shows how the relationship between work and IADL allocation has changed over time. We show that the time allotted to IADLs did not change significantly for men or women despite increasing age and decreasing working hours. 

Similarly, there was no significant difference in resting and sleeping time or media viewing time for men or women. However, the time spent on these activities tended to increase with age (see Figure 2).

## 4. Discussion

### 4.1. Summary of Findings (See Table 2)

The purpose of this study was to longitudinally examine changes in the time allocation of each activity in the post-war baby boomer generation using data from public statistical surveys on living time in Japan. Furthermore, the study aimed to investigate how this generation formed occupational balance.

This study found that the time allocated to work, IADLs, and social participation tended to decrease year by year, whereas resting, sleeping, ADLs, and media viewing tended to increase. This trend was similar for both men and women. Comparing weekdays and Sundays, the hours allocated to work on Sundays decreased, but the hours spent resting, sleeping, and watching media increased. We also found that occupational balance patterns on Sundays were largely unaffected by age.

Notably, Japan’s baby boomer generation was characterized by men and women spending different amounts of time on work and household chores. Women spent more time than men on housework, and the time allotted to IADLs remained unchanged for both men and women despite the passage of life and reduced working hours. This study shows that much of the time spent on work that decreased due to retirement shifted to media viewing time. Furthermore, men’s participation in housework did not change significantly from before retirement. 

“Work-life balance” is a big theme. What this study found is that work and family are not mutually exclusive concepts, and that over time, a separate domain has emerged between work and home activities. As the individuals got older, they had less time to work, but that less time was not devoted to housework. It was suggested that what is important in examining occupational balance is how to maintain the balance of time allocation between activities in daily life.

**Table 2 ijerph-20-04060-t002:** Hypotheses and findings gained in this study.

Hypotheses in This Study		Findings
H1	Over time, less working time leads to more time spent in social activities to supplement social participation.	Social participation tended to decrease year by year.
H2	Over time, the occupational balance shifts into harmony, with time for social activities and time for activities that are meaningful to the individual.	The hours allocated to work on Sundays decreased, but the hours spent resting, sleeping, and watching media increased. Occupational balance patterns on Sundays were largely unaffected.
H3	Over time, women in this generation spend less time on IADLs, such as housework, and more time with family, such as leisure.	Women spent more time than men on housework, and the time allotted to IADLs remained unchanged.
H4	Over time, men in this generation spend less time working and more time doing IADLs such as housework and spending more time on family activities such as leisure.	Men’s participation in housework did not change significantly from before retirement.

### 4.2. The Relationship between Weekdays and Sundays

A lifetime is not formed by just one type of day but by weekdays, Saturdays, and Sundays, then months and years. Occupational balance is not formed immediately, and it takes time for it to become a habit through repeated activities that form occupational patterns. 

The data extracted in this study cover the period when the people of the baby boomer generation were in their 40s to 60s, when working and living activities at home had become habits and living was carried out. However, in the 60s, there was no shift to social activities such as volunteering and community activities due to the reduction in working hours on weekdays due to retirement. However, this situation does not appear to be unique to Japan. Research by Park et al. [23] showed that the role of urban elderly in South Korea is mainly confined within the family. In other words, participation levels were low in the areas of IADL, work, leisure, and social participation. This older age group needs to be provided with opportunities for social activity through wider professional involvement. In Japan, the time allocation of men in their 60s on weekdays is similar to the time allocation on Sundays when they were working. Thus, they might have expected life after retirement to be an extension of Sundays.

As the results showed, neither men or women significantly changed how they spent their Sundays in the 20 years examined by the study. By fixing their Sunday time allocation, they may have attempted to maintain stable weekday routines to cope with unpredictable changes in weekday living activities. 

However, it should be noted that there was a gender difference in the use of Sunday time. Of course, individual differences in habit formation must be further investigated, but it is predicted that the pattern of daily behavior on Sundays will be of great help in considering post-retirement life. Dunn notes that habits can form the basis of many human behaviors and support daily life, but they can also interfere with life satisfaction [31]. Occupational patterns can affect occupational balance over the long term. As this study showed, social participation tended to decrease year by year. Working hours decreased over time, and social participation time decreased accordingly. It is an urgent task to consider how to encourage the working generation to form the habit of engaging in social activities.

The formation of occupational balance is not autonomous, as it reflects the culture and values of that era. Therefore, it is also important to consider the underlying Japanese cultural background and values that create the baby boomer generation’s consciousness.

### 4.3. Gender Differences

The results of this study suggest that women in the baby boomer generation lived stable and well-balanced lives across the examined decades, whereas men showed some changes in life balance in the study period. However, this means that there was no significant change in the time allocation among women. In fact, women spent more time on household chores than men, and the time allotted for the IADL remained the same. Women may have been unable to change their time allocations because of the overlapping roles of ADLs, work, and household activities in their daily lives. Evans et al. [32] named the role balance of working women as the working “sandwich” generation of women. Gender differences in occupational balance need to be analyzed not only from the perspective of time use but also from that of overloaded and conflicting roles in daily life.

By contrast, men showed significant differences in occupational balance on weekdays and Sundays. Matuska [33] stated that “workaholism” can be a form of occupational imbalance, and men of this generation in Japan have been commonly described as “work-centric” or “overworked.” On the other hand, the men were able to avoid devoting time to housework. As shown in the results of this study, men’s participation in housework activities had no effect on retirement.

Achieving occupational balance—and living a well-balanced life—means being able to fulfill the roles required in every aspect of one’s life. However, making a single, fixed role the center of one’s life leads to a fixed balance. We must be aware that it is difficult to maintain a well-balanced daily life when roles are overloaded or limited.

It is important to maintain a good work–life balance regardless of one’s gender. Noda investigated the effect of work–life balance on life satisfaction using data from OECD countries and suggested that institutional design that appropriately incorporates the work–life balance of both men and women is important for increasing life satisfaction [34]. Policymakers may, therefore, benefit from considering the valuable perspective of occupational balance.

### 4.4. Readjusting Occupational Balance

This research focused on the occupational balance of the baby boomer generation between their 40s and 60s, during which they transitioned from work into retirement. Jonsson et al. examined this transition using the narrative slope [35]. Pettican and Prior suggested that the retirement transition is a period of significant readjustment and noted a close relationship between individuals’ engagement in occupation and their perceived health and well-being during this period [36]. People hold different values that are reflected in how they spend their time: at work, with their families, on their hobbies, etc. What is important is the recognition that “I can lead a balanced life.” One must constantly readjust their occupational balance to be flexible in responding to life events.

Occupational imbalance involves both time-based conflicts (e.g., insufficient or overlapping time) and behavior-based conflicts (e.g., single-role or multi-role engagement). Therefore, there is a need for readjustment support to prevent roles from overflowing or being restrained due to these conflicts. Another notable result of this study is that the time spent watching media gradually increased, and working hours were replaced by media viewing rather than social participation. It is certainly important to watch TV without doing anything; in some cases, we may collect information by actively consuming media. According to Yerxa, “People need to develop skills to cope with this meaningless overload by learning to see boredom as a signal of action” [6]. We suspect that many baby boomers suffer from this “meaningless overload.” However, whether viewing media constitutes a meaningful time or a way to kill time may be a function of how one forms their work–life balance before retirement.

It may be necessary to either take action with an awareness of gradual retirement before reaching retirement age or implement a support system for gradual retirement. Clouston states that the current economic system fosters not only a work–life imbalance but also occupational imbalance [37]. Paid work is prioritized as a critical activity in the current social structure and demands most of an individual’s time and energy resources. We suggest that older adults should learn that participating in unpaid work and volunteer activities contributes to their health during periods of stable work (40s to 50s).

### 4.5. Study Limitations and Implications for Practice

This study was based on survey data conducted by a Japanese public agency. Furthermore, because this study was based on a single survey data in Japan, the backgrounds of the participants (e.g., behavioral preferences, psychological attributes, occupation, family structure, etc.) and social background such as employment environment and social security system were not taken into consideration. 

To resolve them, further data from other countries should be collected and considered to see if similar results are obtained in other countries. Additionally, since this study analyzed only one generation (the baby boomer generation), it is necessary to examine whether similar time allocation can be obtained by investigating other generations in chronological order. In contemplating the current situation of aging from multiple perspectives, it is important to examine the common factors that affect the change in work balance over time by comparing countries with different cultures and social systems. 

In addition, it is necessary to examine the effects of living behavior based on time allocation on psychological aspects such as happiness and satisfaction. Though the investigation of occupational balance is important, it is also necessary to consider theorization that integrates behavior and psychology in everyday life. Allen et al. described the challenges of research on work–family conflicts [38], and Jonsson and Persson reported a model of occupational balance analysis based on flow theory analysis [39]. However, Grzywacz and Carlson stated that the systematic theorization of work–life balance has not caught up with the high level of interest it has attracted [40]. Therefore, as Wagman et al. state, systematic research is needed on the level of occupational balance and how to strengthen it [41]. Therefore, this study can be positioned as a part of systematic research.

Future research should focus on the practical application of occupational balance and its theorizing for public policy making. First, we need to further examine how gender differences affect occupational balance. This is also related to the values of work and domestic labor. The relationship between work and family life differs in each country and should be investigated more carefully. It is also expected to be related to economic conditions and family composition. Second, how social security and welfare systems affect this occupational balance may be related to public health issues. In some cases, it is necessary to examine how occupational imbalances arise. Work and social activities provide us with an important role in our lives. Lewis and Lemieux stated that social participation is considered a right and that public policy can work to promote this right [42]. To this end, we must further consider the implications of occupational balance and imbalance.

Furthermore, we may need to consider developing social education programs as a means of practicing occupational balance. As described by Forhan and Backman [12], it is suggested that incorporating the occupational balance theory into social education programs further promotes occupational balance practice. The development of educational programs on this occupational balance will also be useful for policy making.

## 5. Conclusions

The findings of this study showed gender differences in occupational balance among baby boomers in Japan. In addition, looking at changes in time allocation in one generation longitudinally revealed that the readjustment of occupational balance is necessary for periods of transitioning life roles, such as retirement. This study suggests that if this readjustment is not carried out properly, it will lead to the uptake of new activities to kill time, leading to role overload and loss.

## Figures and Tables

**Figure 2 ijerph-20-04060-f002:**
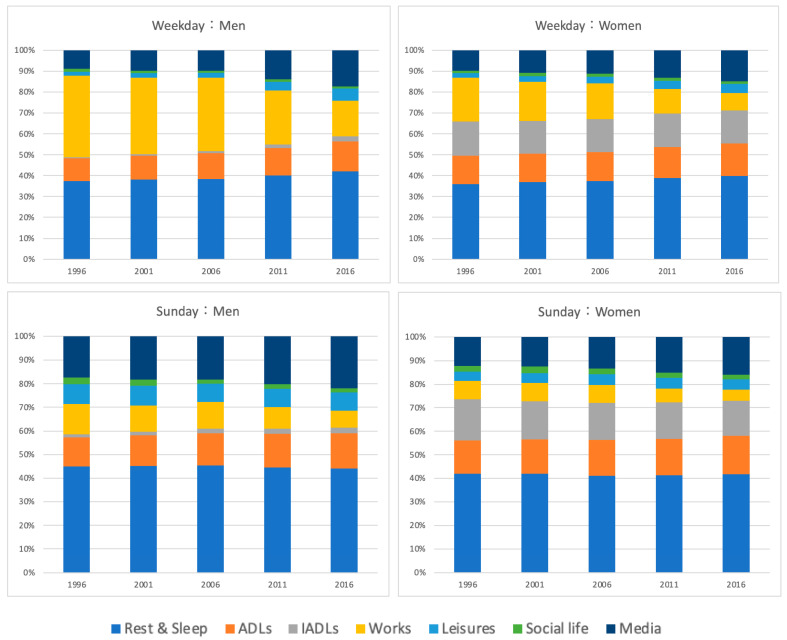
Occupational performance patterns.

**Figure 3 ijerph-20-04060-f003:**
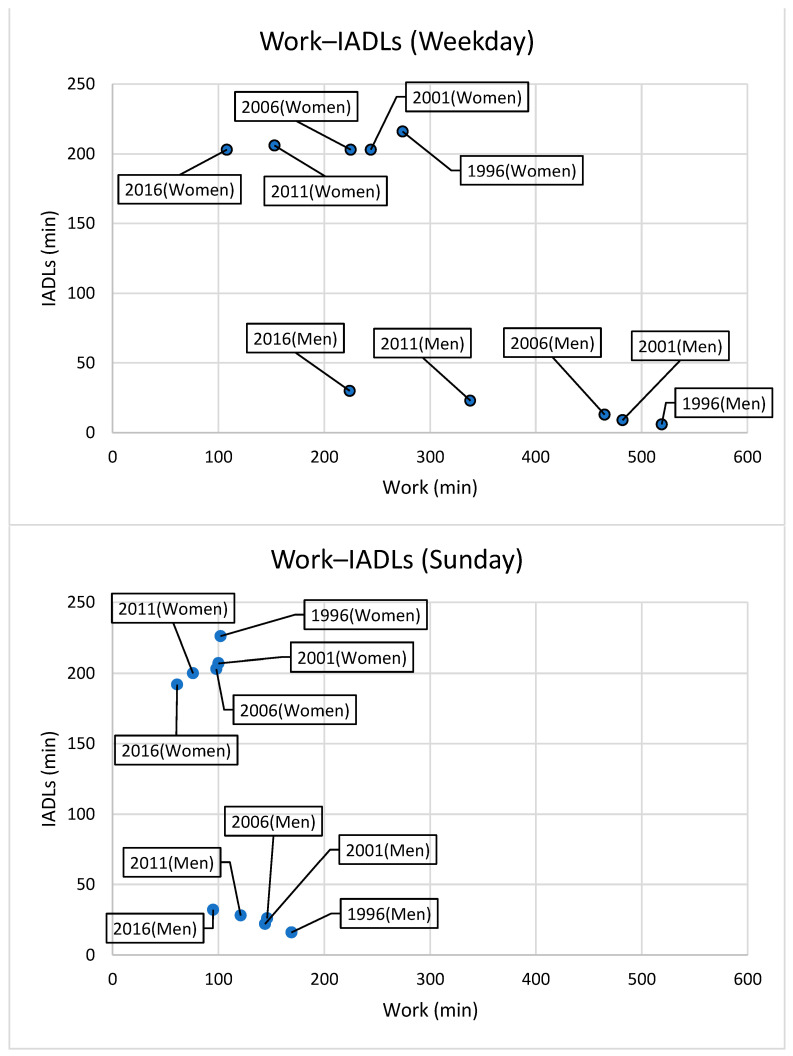
Temporal correlation between work and IADLs (weekdays, Sunday).

**Table 1 ijerph-20-04060-t001:** Characteristics of time allocation for Japan’s baby boomers in every 5 years (1996–2016).

	Women					Men				
Year Age	1996 (45–49)	2001 (50–54)	2006 (55–59)	2011 (60–64)	2016 (65–69)	1996 (45–49)	2001 (50–54)	2006 (55–59)	2011 (60–64)	2016 (65–69)
**Weekday (Monday-Friday)**										
Rest & Sleep	473	482	488	506	515	499	504	509	527	551
ADLs	177	179	181	193	203	147	154	164	176	189
IADLs	216	203	203	206	203	6	9	13	23	30
Works	274	244	225	153	108	519	482	465	338	224
Leisure	28	36	40	51	55	24	30	33	56	77
Social Participation	19	21	19	19	18	20	17	13	15	12
Media Viewing	127	142	146	171	191	118	129	130	184	228
**Weekend (Sunday)**										
Rest & Sleep	545	537	531	534	543	590	589	593	586	577
ADLs	182	188	197	199	211	162	169	178	188	195
IADLs	226	207	203	200	192	16	22	26	28	32
Works	102	100	98	76	61	169	144	146	121	95
Leisure	50	55	60	58	59	109	108	102	101	102
Social Participation	32	34	30	28	23	38	35	25	26	22
Media Viewing	159	161	172	196	208	228	238	237	266	187

Note. Year = The survey year; Age = Age of subjects in the survey year; ADLs = Meal and self-care; IADLs = Housework; Media Viewing: TV, radio, newspaper, and magazine. Source: “Survey on Time Use and Leisure Activities” Statistics Bureau of Japan (1996, 2001, 2006, 2011, and 2016).

## Data Availability

The Statistics Bureau of Japan plays a central role in the official statistical system by producing and disseminating basic official statistics, such as the Population Census. These statistical data can be viewed and used freely.

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
