# Peer review of "Longitudinal Changes in Occupational Balance among Baby Boomers in Japan (1996–2016)"

_ijerph, 2023, doi:10.3390/ijerph20054060_

Round 1

Reviewer 1 Report

The Author has come up with a good study. Quite Interesting, but at the same time, I was wondering how exactly is the data you have collected based on the Survey on Time Use and Leisure Activities published by the Statistics Bureau of Japan. It looks a little confusing regarding data being taken for the exaction. The Author can still do better analyses, adding value to the study. I request the Author to add the Survey on Time Use and Leisure Activities Secondary data as an annexure in the study. The discussion part is good. The Author should include a Review of the literature in this study which I found is missing. 

Author Response

The authors would like to thank the reviewers and editors for their time and effort to review this manuscript. Please consider the attached manuscript that has been revised according to the reviews. In the revised manuscript, edits made based on reviewer’s comments are highlighted in red. Our responses to each reviewer’s comments are as follows:

Reply to Reviewer 1.

Comment 1

I request the Author to add the Survey on Time Use and Leisure Activities Secondary data as an annexure in the study.

Answer to comment 1

We agree with you and have incorporated this. The figure shown in the manuscript was difficult to understand, so Table 1 was recreated and inserted on page 4.

Comment 2

The Author should include a Review of the literature in this study which I found is missing.

Answer to comment 2

Thank you for your suggestion. A review of the literature was inserted in the second half of the introduction (page 2, lines 60-75).

We have removed "4.6. Clinical implications". Accordingly, references [31], [34], and [35] have been deleted and reference numbers have been changed.

Again, thank you for giving us the opportunity to strengthen our manuscript with your valuable comments.

Reviewer 2 Report

I found the paper easy to read and follow. However, I feel that significant work must be done towards its improvement. My more specific comments and suggestions follow:

1.) The paper’s Introduction lacks a clear focus. In essence, it should lead to the formulation of a specific Research Question, that needs to be clearly stated at the end of the Introduction. All the content in the Introduction must effectively lead to or support this research question, that will back up the rationale of the existence and importance of conducting this research in the first place (to answer to this research question identified from insight collected from existing literature).

2.) The paper lacks a Background section (between the Introduction and Materials and Methods). In essence, what needs to be added is existing insight from the literature on occupational balance, as well as the parameters connected to it (that affect or are affected by it) in more detail, especially focusing on the parameters researched in this paper (i.e., age, gender, retirement). Once this necessary background has been compiled and presented, the authors need to then formulate specific hypotheses in connection to the research question (that they will have added in the introduction as per my previous comment). These research hypotheses are then to be proven/discarded through their more detailed analysis (as they explain it in the Materials & Methods section).

3.) I suggest that the authors revise the structure of the materials & methods section. It is split into many sub-chapters that can be merged into less sub-chapters. Moreover, “Designs” talks about the “Research Design” of the study and should be accompanied by a Figure, where the authors outline the research steps they take in this paper (the process followed). The subsections “Data Source”, “Data Collection”, and “Data Analysis” should be revised to be merged into a more cohesive narrative of the process followed to collect and analyze the data.

4.) In the last paragraph of the section on “Data analysis”, the authors state that “In addition, this study examined whether the time allocation is affected by the day of the week (weekday/Sunday), because working hours are expected to be different on weekdays and Sundays, and gender, because the social advancement of women in the baby boomer generation was still insufficient.”. All these facts are examples of the content that should have been supported in the missing “Background” section by existing literature – and related hypotheses should have been formulated. Otherwise (at present) they are not supported and solely reflect the authors’ personal views.

5.) The results section should start off with an analysis of the sample (Sample Characteristics), where the authors can explain the demographic mix of participants. Some of this information is for some reason at present located in section 2.3 (data collection). They should be moved here and further details on the demographics be added.

6.) The sub-sections of the results section contain just a few lines each. They should be further enriched and also merged if needed. For example, provide some indicative numerical information also where you generally say “has been increased” – for example, by what percentage? what rate of increase? etc.

7.) Table 1 looks like has been copied/pasted from another source. If the content is original and has been built by the authors, they should insert an editable Table here with larger fonts (according to MDPI specifications). If not, simply reproducing a published table (especially without license and no citation) would constitute plagiarism. The same stands for Figure 2. The figure itself, as well as its caption have been copied-pasted from another document/source.

8.) The discussion and conclusion of the paper must be revised according to the changes that will be made to the paper according to the previous comments. The specific research question and hypotheses should be utilized to (re)build it accordingly. Moreover, insight from the Background section (that the authors add) should also be utilized to further enrich the comparison of existing insight from the literature to the findings in this paper (e.g. in section 4.2. no comparison is made to existing literature – it looks like a plain summary of findings). I suggest that the authors also build a table in the discussion that repeats the hypotheses (that they will formulate), and summarizes the insight acquired with regards to each one from both the existing literature, as well as the findings in this research.

9.) In addition to the above, the authors need to pay special attention into shedding much more light to the theoretical and practical (not just clinical) contribution of their work. This is most important in this case, as they have not conducted any data collection (existing data from a third source is re-examined), and they have not added any other means of verification (e.g., experimentation or in-depth interviews). Hence, they need to offer stronger justification for the merits of their work both for researchers and practitioners. Otherwise, the paper would lack rationale and/or specific theoretical and practical value.

10.) The study limitations have not adequately been explained. Apart from the fact that this study was based on a survey (e.g., you also need to talk about single-source bias), more should be told. For example, that the survey was only performed in one country (Japan), that only one batch/generation of elders was analyzed (the analysis of other batches of the same age group in the future could bring forth similarities or differences), etc. After adequately reflecting on the limitations of this study, the authors also need to accordingly enrich their suggestions for future work (that would perhaps cover these limitations in the future, or lead to richer results).

11.) The conclusion could be merged with the discussion, as it contains more or less similar information.

Author Response

The authors would like to thank the reviewers and editors for their time and effort to review this manuscript. Please consider the attached manuscript that has been revised according to the reviews. In the revised manuscript, edits made based on reviewer’s comments are highlighted in red. Our responses to each reviewer’s comments are as follows:

Reply to Reviewer 2.

Comment 1

1.) The paper’s Introduction lacks a clear focus. In essence, it should lead to the formulation of a specific Research Question, that needs to be clearly stated at the end of the Introduction. All the content in the Introduction must effectively lead to or support this research question, that will back up the rationale of the existence and importance of conducting this research in the first place (to answer to this research question identified from insight collected from existing literature).

Answer to comment 1

We agree with your helpful comment. As you suggested, we inserted the research question at the end of the "Introduction" (page 2, lines 77-83).

Comment 2

2.) The paper lacks a Background section (between the Introduction and Materials and Methods). In essence, what needs to be added is existing insight from the literature on occupational balance, as well as the parameters connected to it (that affect or are affected by it) in more detail, especially focusing on the parameters researched in this paper (i.e., age, gender, retirement). Once this necessary background has been compiled and presented, the authors need to then formulate specific hypotheses in connection to the research question (that they will have added in the introduction as per my previous comment). These research hypotheses are then to be proven/discarded through their more detailed analysis (as they explain it in the Materials & Methods section).

Answer to comment 2

We agree with your helpful comment. We combined comments 1 and 2 and inserted them at the end of the "Introduction" (page 2, lines 77-83). In accordance with this, the “Materials and Methods” were revised.

Comment 3

3.) I suggest that the authors revise the structure of the materials & methods section. It is split into many sub-chapters that can be merged into less sub-chapters. Moreover, “Designs” talks about the “Research Design” of the study and should be accompanied by a Figure, where the authors outline the research steps they take in this paper (the process followed). The subsections “Data Source”, “Data Collection”, and “Data Analysis” should be revised to be merged into a more cohesive narrative of the process followed to collect and analyze the data.

Answer to comment 3

We agree with your suggestion. "Materials and Methods" changed chapter structure. Figure 1 (page 3) created newly and inserted to clarify the study procedure. Also, the “Study designs” (pages 2-3) and “Procedures for data analysis” (page 3-4) were described to clarify.

Comment 4

4.) In the last paragraph of the section on “Data analysis”, the authors state that “In addition, this study examined whether the time allocation is affected by the day of the week (weekday/Sunday), because working hours are expected to be different on weekdays and Sundays, and gender, because the social advancement of women in the baby boomer generation was still insufficient.”. All these facts are examples of the content that should have been supported in the missing “Background” section by existing literature – and related hypotheses should have been formulated. Otherwise (at present) they are not supported and solely reflect the authors’ personal views.

Answer to comment 4

Thank you for your suggestion. The following sentence (“In addition, this study examined whether the time allocation is affected by the day of the week (weekday/Sunday), because working hours are expected to be different on weekdays and Sundays, and gender, because the social advancement of women in the baby boomer generation was still insufficient.”) was deleted. It was described as a review of the literature in the second half of the "Introduction" (page  2, lines 64-75).

Comment 5

5.) The results section should start off with an analysis of the sample (Sample Characteristics), where the authors can explain the demographic mix of participants. Some of this information is for some reason at present located in section 2.3 (data collection). They should be moved here and further details on the demographics be added.

Answer to comment 5

We appreciate your helpful comment and rewrote “Results”. We inserted the characteristics of the data sample (3.1.) at the beginning of the results (page 4, lines 143-152).

Comment 6

6.) The sub-sections of the results section contain just a few lines each. They should be further enriched and also merged if needed. For example, provide some indicative numerical information also where you generally say “has been increased” – for example, by what percentage? what rate of increase? etc.

Answer to comment 6

We agree with you. We inserted numerical data in the results (pages 5-7).

Comment 7

7.) Table 1 looks like has been copied/pasted from another source. If the content is original and has been built by the authors, they should insert an editable Table here with larger fonts (according to MDPI specifications). If not, simply reproducing a published table (especially without license and no citation) would constitute plagiarism. The same stands for Figure 2. The figure itself, as well as its caption have been copied-pasted from another document/source.

Answer to comment 7

Thank you for your suggestion. Figures and tables have been recreated and inserted (Table 1; page 4, Figure 2; page 5, Figure 3;page 6). These figures and tables are original.

Comment 8

8.) The discussion and conclusion of the paper must be revised according to the changes that will be made to the paper according to the previous comments. The specific research question and hypotheses should be utilized to (re)build it accordingly. Moreover, insight from the Background section (that the authors add) should also be utilized to further enrich the comparison of existing insight from the literature to the findings in this paper (e.g. in section 4.2. no comparison is made to existing literature – it looks like a plain summary of findings). I suggest that the authors also build a table in the discussion that repeats the hypotheses (that they will formulate), and summarizes the insight acquired with regards to each one from both the existing literature, as well as the findings in this research.

Answer to comment 8

We agree with you. We revised “4.1.Summary of finding” (page 7, lines 264-287) and “4.2.The relationship between weekdays and Sundays” (page 7, lines 289-312) .

Comment 9

9.) In addition to the above, the authors need to pay special attention into shedding much more light to the theoretical and practical (not just clinical) contribution of their work. This is most important in this case, as they have not conducted any data collection (existing data from a third source is re-examined), and they have not added any other means of verification (e.g., experimentation or in-depth interviews). Hence, they need to offer stronger justification for the merits of their work both for researchers and practitioners. Otherwise, the paper would lack rationale and/or specific theoretical and practical value.

Answer to comment 9

We agree with you. We have reflected this comment (pages 9-10, lines 389-405). We have removed "4.6. Clinical implications". Accordingly, references [31], [34], and [35] have been deleted and reference numbers have been changed.

Comment 10

10.) The study limitations have not adequately been explained. Apart from the fact that this study was based on a survey (e.g., you also need to talk about single-source bias), more should be told. For example, that the survey was only performed in one country (Japan), that only one batch/generation of elders was analyzed (the analysis of other batches of the same age group in the future could bring forth similarities or differences), etc. After adequately reflecting on the limitations of this study, the authors also need to accordingly enrich their suggestions for future work (that would perhaps cover these limitations in the future, or lead to richer results).

Answer to comment 10

We agree with you. We have reflected this comment (page 9, lines 368-383).

Comment 11

11.) The conclusion could be merged with the discussion, as it contains more or less similar information.

Answer to comment 11

Thank you for your suggestion. We have removed unnecessary parts as "Conclusion". I decided to state it briefly as the conclusion of this study (page 10, lines 407-412).

Again, thank you for giving us the opportunity to strengthen our manuscript with your valuable comments.

Round 2

Reviewer 1 Report

Dear Authors,

I have reviewed the revised manuscript titled "Longitudinal changes in occupational balance among baby boomers in Japan (1996-2016)" and would like to commend you on the improvements made in response to the reviewer comments. The manuscript is well-written, and the findings provide valuable insights into the time allocation patterns of the baby boomer generation in Japan and the gender differences in occupational balance.

I appreciate the attention given to the feedback provided, particularly with regards to the clarity of the research questions and the presentation of the results. The revised manuscript is much more concise and easy to follow, with clear headings and subheadings that guide the reader through the research design, methods, and findings.

Overall, the study contributes to the understanding of the challenges that individuals face when transitioning into retirement, and highlights the importance of readjusting occupational balance during life role changes. I believe this manuscript will be of interest to a broad readership, including researchers and practitioners interested in occupational health and gerontology.

Thank you for the opportunity to review the revised manuscript.

Author Response

Reply to Reviewer 1.

Thank you for providing important insights.

Through peer review of this research, I was able to identify research themes that should be examined in more detail.

Again, thank you for taking the time and energy to help us improve the paper.

Reviewer 2 Report

Although some work has been put to improving the paper, unfortunately some of my important comments still remain unanswered from the previous round. Hence, the paper has not been adequately improved. I would like to once more turn the attention of the authors to the following comments:

2.) The paper lacks a Background section (between the Introduction and Materials and Methods). In essence, what needs to be added is existing insight from the literature on occupational balance, as well as the parameters connected to it (that affect or are affected by it) in more detail, especially focusing on the parameters researched in this paper (i.e., age, gender, retirement). Once this necessary background has been compiled and presented, the authors need to then formulate specific hypotheses in connection to the research question (that they will have added in the introduction as per my previous comment). These research hypotheses are then to be proven/discarded through their more detailed analysis (as they explain it in the Materials & Methods section).

The background section is essential in order to in turn formulate specific hypotheses. This is imperative in my view for this study.

4.) In the last paragraph of the section on “Data analysis”, the authors state that “In addition, this study examined whether the time allocation is affected by the day of the week (weekday/Sunday), because working hours are expected to be different on weekdays and Sundays, and gender, because the social advancement of women in the baby boomer generation was still insufficient.”. All these facts are examples of the content that should have been supported in the missing “Background” section by existing literature – and related hypotheses should have been formulated. Otherwise (at present) they are not supported and solely reflect the authors’ personal views.

No literature has been cited in the few lines that have been added to the introduction by the authors. No background section has been added. Just a paragraph added based on the authors' own beliefs. That is no way to support specific hypotheses. Please revert to my previous round comments 2 & 4. Significant work must be put into adding a whole new section containing insight from the literature on the research background. According to this background specific Hypotheses must be formulated and stated formally (i.e. Hypothesis H1: Parameter X will affect parameter Y in that way, etc.).

The discussion must be revised according to the hypotheses that will be formulated based on previous round comments also (see my previous round comment #8.

Author Response

The author would like to thank the reviewers and editors for their time and effort to review this manuscript. Please consider the attached manuscript that has been revised according to the reviews. In the revised manuscript, edits made based on Reviewer2’s comments are described in red.

Reply to Reviewer 2.

Comment 1

Although some work has been put to improving the paper, unfortunately some of my important comments still remain unanswered from the previous round. Hence, the paper has not been adequately improved.

The background section is essential in order to in turn formulate specific hypotheses. This is imperative in my view for this study.

Answer to comment 1

We agree with your helpful comment. As you suggested, we have restructured the "introduction" (pages 1-3).

1.1. Baby boomer issues in Japan

1.2. Theoretical background

1.3. Hypothesis development

Comment 2

No literature has been cited in the few lines that have been added to the introduction by the authors. No background section has been added. Just a paragraph added based on the authors' own beliefs. That is no way to support specific hypotheses. Please revert to my previous round comments 2 & 4. Significant work must be put into adding a whole new section containing insight from the literature on the research background. According to this background specific Hypotheses must be formulated and stated formally (i.e. Hypothesis H1: Parameter X will affect parameter Y in that way, etc.).

Answer to comment 2

We agree with your helpful comment. In particular, in "1.3. Hypothesis development", I added references ([22], [23], and [24]) and four hypotheses (pages 2-3).

Comment 3

The discussion must be revised according to the hypotheses that will be formulated based on previous round comments also (see my previous round comment #8.)

Answer to comment 3

We agree with your suggestion and amended the "Discussion". We created and inserted Table 2 at the end of “4.1. Summary of findings” (page 8). Also, we inserted reference [23] in “4.2. The relationship between weekdays and Sundays” and added a discussion (page 13, lines 335-342 and 351-358), and, we added considerations in “4.3. Gender differences”. Finally, we deleted the subheading "4.5. Meaningful time" and integrated the discussion as "4.4. Readjusting occupational balance" (page 10).

Reference numbers have been revised due to newly added references.

Again, we appreciate all your insightful comments. Thank you for taking the time and energy to help us improve the paper.